# Cage-Free Pullets Minimally Affected by Stocking Density Stressors

**DOI:** 10.3390/ani14101513

**Published:** 2024-05-20

**Authors:** Meagan E. Abraham, Cara I. Robison, Priscila B. S. Serpa, Natalia J. Strandberg, Marisa A. Erasmus, Gregory S. Fraley, Gisela F. Erf, Darrin M. Karcher

**Affiliations:** 1Department of Animal Sciences, Purdue University, West Lafayette, IN 47907-2050, USA; abraha27@purdue.edu (M.E.A.); merasmus@purdue.edu (M.A.E.); gfraley@purdue.edu (G.S.F.); 2Department of Animal Sciences, Michigan State University, East Lansing, MI 48824-2604, USA; oconn107@msu.edu; 3Department of Biomedical Sciences and Pathobiology, VA-MD College of Veterinary Medicine, Blacksburg, VA 24061, USA; pserpa@vt.edu (P.B.S.S.); nstrand@vt.edu (N.J.S.); 4Department of Poultry Science, System Division of Agriculture, University of Arkansas, Fayetteville, AR 72701, USA; gferf@uark.edu

**Keywords:** pullet, layer hen, stocking density, welfare, stress

## Abstract

**Simple Summary:**

Simple Summary: The first 16 weeks of life for a laying hen is the pullet phase. Experiences and management during this phase are critical for the long-term success of a hen, but few studies have evaluated pullet management in cage-free systems. This study evaluated the effects of two density or space allotments and two pullet strains in cage-free systems. Bird condition, physiology, immunology, and production parameters were evaluated. Stocking density only affected the size of one immune organ, the bursa of Fabricius. The relative bursal weight was higher in the low-density group. The feed conversion rate was improved in the low-density group for both strains. The brown strain had decreased uniformity and worse tail and total feather coverage at the high-stocking density. The white strain had improved uniformity and worse tail and total feather coverage at the low-stocking density. The majority of parameters evaluated had strain and age main and/or interaction effects only. Ultimately, cage-free pullets had limited negative effects at the high and low-stocking densities used.

**Abstract:**

Management choices during the pullet phase can affect behavior, welfare, and health later in life, but few studies have evaluated the pullet phase, particularly in extensive housing systems. This study was a 2 × 2 factorial randomized complete block design (RCBD) with two strains and two stocking densities. The Lohmann LB-Lite and Lohmann LSL-Lite were housed on the floor at high-stocking density (619–670 cm^2^/bird) and low-stocking density (1249–1352 cm^2^/bird), which changed with age from 2 to 16 weeks of age (WOA). Bird-based measures of appearance, blood parameters, organ measurements, and production values were evaluated. Stocking density alone affected (*p* < 0.05) only relative bursal weight (% of body weight)—3.32% in the low-density versus 3.08% in the high-density group. High-stocking density was correlated with decreased uniformity (high—89.33 ± 0.24%; low—90.41 ± 0.24; *p* < 0.02) and worse feather coverage in the brown strain. High-stocking density was correlated with greater uniformity (High—90.39 ± 0.24%; Low—88.47 ± 0.24%; *p* < 0.001) and better feather coverage in the white strain. This study’s feed conversion ratio (FCR) was improved by 0.07 in the low-stocking density for both strains. The remaining parameters were affected by strain and age only. Thus, while stocking density effects vary slightly depending on the strain used, cage-free pullets had limited negative effects at both the high and low-stocking densities tested in this study; there were few to no changes in the numerous bird-based welfare parameters tested.

## 1. Introduction

The laying hen industry in the United States consists of approximately 330–340 million laying hens [1] that must first go through the growing phase known as the pullet phase. The pullet phase accounts for approximately 25% of a laying hen’s lifetime [2] and encompasses approximately 0–16 weeks of age (WOA). During this time, the pullet reaches sexual maturity and grows to its full skeletal capacity. Early experiences during the pullet phase, including housing and environment, can affect the health, behavior, and welfare of the birds for the duration of their lives [3,4,5,6]. Therefore, proper pullet rearing environments are crucial to the later adaptation of an adult hen to her housing environment [4,5,6]. Pullets are presently raised in all types of housing systems, such as conventional cages, furnished cages, aviaries, and floor-based systems [7], with conventional cages being the most prevalent type of laying hen housing across the industry [8,9] and the most used system for pullet rearing [5]. However, as the poultry industry moves toward more extensive production, there is a greater need to understand how pullets adapt to cage-free housing systems and how to best manage pullets in these cage-free environments. 

An important component of management is stocking density or the number of birds placed in a given space. Stocking density affects a chicken’s ability to access resources and exhibit natural behaviors [8,10], and chickens stocked too densely will have negative health and production outcomes [11]. Additionally, stocking density is cited as one of the most important factors when determining an animal’s well-being by both consumers and farmers [12,13], and farmers believe that product quality increases at lower stocking densities [12]. For many decades, however, stocking density studies in pullets have been few and far between [3,14], and legislation has largely ignored pullet stocking density [2]. Most cage-free stocking density guidelines have been based on either the number of adult hens per area or the total live weight of the birds, as is common in broilers [2]. Neither of these methods accounts for pullets that are growing slower than broilers but faster than adult hens. An ideal stocking density for cage-free pullets has not been established, so producers often use management guidelines generated by the company that provided the chicks. These guidelines are useful but are based mainly on anecdotal evidence [8].

Utilized cage-free stocking densities vary widely; commercial pullet growers in the United States typically use densities ranging from 413 to 929 cm^2^/bird (personal communications), while commercial Swiss producers have reported average stocking densities of 763–781 cm^2^/bird [15]. In research settings, there is an even wider spread of densities recommended—464–1335 cm^2^/bird [16]—and computer-based modeling recommends 667–1111 cm^2^/bird [2] or 714–909 cm^2^/bird [17]. Previous studies of cage-free pullets have tested anything from 348 to 5882 cm^2^/bird with varying effects on behavior, production, heterophil to lymphocyte (H:L) ratios, corticosterone, bird condition, and immunity [6,14,18,19,20,21,22,23,24,25,26]. However, pullet strains, and, therefore, body sizes and behaviors, differ among studies, and no conclusive results can be drawn from these studies that would identify optimal stocking density, particularly in a cage-free system. Further work is needed to identify optimal densities for different pullet strains and housing types.

There are multiple ways to evaluate a bird’s response to different stocking densities. The earliest welfare research focused largely on resource-based measures of welfare and asked the question of whether the birds had enough access to feed, water, etc. [27]. However, animal-based welfare parameters represent an additional route to identify optimal density. Animal-based parameters are assessments of the animal itself, which examine how the animal perceives their environment by evaluating behavior, physiology, immunology, and production outputs. The development and validation of bird-based welfare parameters is not simple; however, past research attempting to do so has generated mixed results that appear partially dependent on, among other things, the stressor applied, the strain of bird evaluated, and the housing environment [28,29,30,31,32,33].

Not surprisingly, these bird-based parameters have not been tested frequently in pullets, and their validation status as markers of welfare is still in question. Stress and poor welfare are not necessarily synonymous, but it is believed that welfare generally decreases during periods of stress [34], especially during periods of chronic stress [35]. To validate parameters of welfare and stress in pullets, this study sought to identify changes in physiology, immunocompetence, production, and outward appearance of pullets housed at one of two stocking densities. The hypothesis was that pullets housed at a higher stocking density would have higher levels of stress that would manifest as greater alterations from baseline for measurements of physiology, immunocompetence, production, and outward appearance. 

## 2. Materials and Methods

### 2.1. Study Design and Housing

This study was a 2 × 2 factorial randomized complete block design (RCBD) with two strains, Lohmann LB-Lite (Brown) and Lohmann LSL-Lite (White), and two starting stocking densities of 619 (high-stocking density; HSD) and 1249 (low-stocking density, LSD) cm^2^/bird. The project and procedures were reviewed and approved by the Purdue University Animal Care and Use Committee (Protocol#: 1908001938). From 1–7 days of age, 2930 beak-trimmed chicks (1465 of each strain) were placed in 3.05 × 2.44 m litter floor pens. On day 8, chicks were allocated their final litter floor pens across three rooms, 59 chicks/pen, generating 12 replicate pens for each stocking density × strain combination. Each pen had one bell drinker and two bell feeders, providing 7.06 cm/bird of feeder space and 3.53 cm/bird of drinker space. At 3, 6, and 12 WOA, 10, 20, and 20 birds per treatment, respectively, were sampled, generating the stocking density and resource allocations detailed in Table 1. A lighting protocol and feeding program based on the breed management guide [36] was followed with one exception. The light was reduced to approximately 1 footcandle from day 96 to the termination of this study due to moderate to severe feather pecking over the rump and tail areas of birds in four pens that spanned both strains and stocking densities used. In brief, the lighting program was as follows: birds were provided intermittent lighting (4 h of light, 2 h of darkness for 4 total cycles per day) from days 1 and 2 (10 footcandles) and from days 3–13 (5 footcandles). From days 14–20, birds were provided 19 h of light (3 footcandles); from days 21–27, birds were provided 18 h of light (2.5 footcandles), and from day 28 on, an hour of light was reduced each week until day 70 when birds received 11 h of light. Birds received 11 h of light from day 70 to the termination of this study (1.5–2 footcandles maintained for days 28–111; 2–2.5 footcandles maintained for days 112–119). 

Body weights were taken prior to feeding on 10 birds/pen weekly, five tagged and five untagged birds. The same five tagged birds/pen were followed for the duration of this study and were used for blood collection. Body weight gain was calculated by subtracting average bird weights in a pen at the start of this study from average bird weights at the end of this study. Pullets were fed 3 different diets: a “starter” diet from 1–3 WOA; a “grower” diet from 4–8 WOA; and a “developer” diet from 9–16 WOA. Feed consumption was measured weekly. The study feed conversion rate (FCR) was calculated by taking total feed consumption throughout this study and dividing it by the estimated pen body weight gained. Mortality and room temperature (°F) were recorded daily. Room temperatures were fine-tuned as needed with individual room thermostat settings. Pullets were vaccinated, as would be common within commercial flocks and as reported by Gast et al. [37]. Birds received the Newcastle vaccine as a coarse spray on days 13, 33, and 56 and as a breast injection on day 84. 

### 2.2. Bird Condition and Welfare Assessment

A welfare assessment was completed at 4, 8, 12, and 16 WOA, including fracture and deviation of the keel bone through physical palpation, footpad condition, plumage damage, and plumage coverage (N = 32/treatment, Table 2). All but feather coverage, feather damage, and shank length were adapted from the Welfare Quality^®^ Assessment Protocol for Poultry [38]. Feather coverage scores were adapted from Morrisey et al. [39] and Arrazola et al. [40]. Individual feather regions scored included the neck, back, tail, belly/vent regions, wings, and legs. Total feather coverage scores were generated from the sum of all 6 body regions; a score of 0 equates to perfect feather coverage, while a total score of 30 is the worst possible score, indicating severe feather loss and skin damage on all body regions. The same observer completed feather coverage and welfare assessments on all birds across all time points to reduce observer variability. The shank length was measured using a digital caliper (Husky 1467H, Home Depot, Atlanta, GA, USA) on the left leg from the hock to the footpad. Shank length was used as a proxy for tarsometatarsal length [41], indicating skeletal size and growth [42,43,44]. 

### 2.3. Organ Collection

At 1 and 6 WOA, 10 birds/treatment were sampled; at 3 and 12 WOA, 20 birds/treatment were sampled, and at 16 WOA, 32 birds/treatment were sampled following euthanasia via cervical dislocation. All birds were weighed before organ removal. The liver, bursa of Fabricius (bursa), and spleen were removed at 1, 3, 6, 12, and 16 WOA, and the 2nd thymus lobe of both right and left sides were removed at 1, 6, 12, and 16 WOA. If multiple thymus lobes were laying on top of each other in the same “chain”, the most medial and ventral 2nd lobe was taken. Organs were collected, imaged, and weighed. The thymus lobes and spleen had measurements of length, width, and height, and the bursa had length and width measurements taken using the software ImageJ^®^ (Rasband. NIH, Bethseda, MD, USA). Additionally, the bursal size was recorded in situ using a bursometer (Boehringer Ingelheim, Ingelheim am Rhein, Germany). Bursas that were in between two sizes on the bursometer were rounded up to the larger size. Organ weight was converted from g to kg and was divided by the bird weight (kg) to generate relative organ weights (%) for the right and left thymus, spleen, bursa, and liver.

### 2.4. Blood Collection and Analysis

At the end of week 2, five birds in all pens received wing bands (Style 898; National Band and Tag Company, Newport, KY, USA) and were colored with fluorescent pink PrimaGlo livestock paint (QC Supply, Schuyler, NE, USA), similar to Arrazola et al.’s study [40], to track individual birds for the duration of this study. Any lost tagged birds were replaced with new tagged birds and followed for the remainder of this study. The untagged birds in the pen were colored with fluorescent purple PrimaGlo livestock paint (QC Supply, Schuyler, NE, USA) to limit any behavioral differences between birds. Paint was initially placed between the shoulders but was moved to the rump and tail areas at later reapplication. The paint was reapplied every two weeks or as needed.

Heparinized syringes [45] with 20-gauge needles were used to collect blood from the brachial vein at 6, 9, 12, and 16 WOA (40, 61, 81, and 110 days of age; n = 20/treatment). Blood was collected from birds within two minutes of being caught to reduce artificial increases in corticosterone from handling stress [46]. Samples with macroscopic evidence of clotting were not evaluated for WBC differential or H:L ratios. At all time points, blood smears of samples were freshly prepared and stained with a Modified Wright’s stain (Hematek stainer, Siemens, Washington, DC, USA). Two trained clinical pathologists performed a 100-cell leukocyte differential count (blinded to experimental conditions), and the heterophil to lymphocyte ratio (H:L) was calculated from the division of the mean percentage of heterophils by the mean percentage of lymphocytes identified. At 9, 12, and 16 WOA, packed cell volume (PCV) was additionally obtained via the microhematocrit method (11,330 rpm for 3 min, Sorvall Microhematocrit centrifuge, ThermoFisher Scientific, Waltham, MA, USA).

All blood samples were then centrifuged at 10,000 rpm for 5 min, and plasma was collected and stored at −20 °C until further analysis. Plasma was analyzed for cholesterol using the Amplex™ Red Cholesterol fluorometric assay (Invitrogen™, Waltham, MA, USA) (n = 20/treatment/age). Samples were diluted 1:250, and a standard curve consisted of 0, 0.5, 1, 2, 4, 6, and 8 μg/mL cholesterol. Plasma was evaluated for the Newcastle Virus IgG/IgY antibody titers using the CK116 NDV indirect ELISA (Biochek, Scarborough, ME, USA) following kit specifications (n = 20/treatment/age). Corticosterone and cortisol were measured in plasma using an Agilent 1260 Rapid Resolution liquid chromatography system coupled to an Agilent 6470 series QQQ mass spectrometer (LC-MS/MS; Agilent Technologies, Santa Clara, CA, USA) in a method previously described by Tetel et al. (2022; n = 5/treatment; 6 and 16 WOA only).

### 2.5. Statistical Analysis

Linear models of fixed effects, linear mixed models, and multinomial logistic regression were used to analyze the parameters within this study (Table 3) using R^®^ (R Studio^®^, 2021). The main and interaction effects of strain, stocking density, and age were tested. Random effects included a bird, pen, and room, as well as nesting bird within the pen and pen within the room. Models were transformed for normalcy as needed. AIC was the determining factor when selecting models in multinomial logistic regression. Tukey’s post hoc test for multiple comparisons was used to compare means within a group, and Pearson’s correlations were used to generate correlations. Least squared means ± standard error is presented for all data unless otherwise noted. Statistical significance was defined as *p* < 0.05. 

## 3. Results

### 3.1. Bird Condition

There was no effect of age, stocking density, or strain on footpad scores, keel tip fractures, or keel scores in this study. There were no footpad or keel scores above zero, and there were no keel tip fractures present. Additionally, neck and leg feather coverages were not impacted for the duration of this study. 

There were minimal changes to feather coverage; about 60% of the birds evaluated throughout this study had no damage to their feathers, and ~34% of the birds evaluated had <50% of feathers missing, damaged, or broken (no tissue damage); scores 1 and 2. There were density × strain interaction effects on tail feather coverage and total feather scores (Table 4) with opposite stocking density effects between strains. In White, the LSD generated worse tail and total feather coverage scores; approximately 40% of the LSD birds scored ≥1 versus only 35% of the HSD birds. For tail feather coverage, White LSD had 32% ≥ 1 scores, and White HSD had 30% ≥ 1 scores. In Brown, the LSD generated better, lower total feather coverage scores; approximately 43% of the LSD birds scored ≥1 versus 48% of the HSD birds. Tail feather coverage scores of ≥1 in Brown were 40% for LSD and 44% for HSD. The greatest feather damage was visible at 16 WOA.

Age, strain, and strain × age affected feather coverage. Feather coverage scores varied throughout this study. At the first time point, 4 WOA, feather coverage was perfect across all regions. Wing coverage was worst at 8 WOA, while back coverage was worst at 12 WOA (Figure 1A). Both tail and total feather scores increased at 8 WOA, but they were at their highest or lowest at 16 WOA (Figure 1A). White sustained less feather damage during the second and third months of this study but had the worst feather coverage at 4 months of age. In addition, White had a lower percentage of high-severity feather coverage scores in the back and wing regions and a higher percentage of perfect tail and total feather coverage scores (Figure 1B). 

### 3.2. Production and Growth Parameters

There were strain × age (*p* < 0.0001), strain × density (*p* < 0.04), density × age (*p* < 0.001), and strain × density × age (*p* < 0.0001) effects on body weight (Figure 2 and Figure A1). Brown was heavier than White at all time points, with differences observed from 7 to 16 WOA (*p* < 0.003). When evaluating the three-way interaction, differences between densities were insignificant within a given strain and age.

Body weight uniformity had strain × age (*p* < 0.002) effects (Figure A2, Appendix A). Uniformity trends were different between the first half and the second half of this study between strains. Brown had greater uniformity for the first 8 WOA, and White had better uniformity at 9, 11, 12, and 14–16 WOA (*p* > 0.05). There were stocking density × strain (*p* < 0.0001) effects on uniformity, and differences between stocking densities within a strain were significant (*p* < 0.02). Uniformity was improved for LSD in Brown (90.41 ± 0.24% vs. 89.33 ± 0.24%; *p* < 0.02) and improved for HSD in White (90.39 ± 0.24% vs. 88.47 ± 0.24%; *p* < 0.0001).

Shank length, a proxy for skeletal size and growth, had strain × age (*p* < 0.0002) effects. Shank length increased from 40.9 ± 0.29 mm at 4 WOA to 79.2 ± 0.29 mm at 12 WOA. After 12 WOA, shank length plateaued and showed limited growth (*p* > 0.05). Unlike body weights, the average shank length was longer in White (66.9 ± 0.31 mm) than Brown (64.4 ± 0.32 mm). White had larger/longer shank length at 4, 8, 12, and 16 WOA compared to Brown (Figure 3), with differences at 8 and 12 WOA (*p* < 0.0001) only. Shank length and body weights were highly correlated, having a r^2^ of 0.915 (Figure A3). Shank length also had stocking density × age effects (*p* < 0.05). However, while LSD had a longer shank length at 4, 8, and 16 WOA, this was not different from HSD (*p* > 0.05; Figure 3). 

Total study feed consumption per pen averaged 274.92 kg and was unaffected by strain or stocking density. Only strain affected the feed conversion ratio (FCR) of the parameters tested. FCR was improved for Brown (3.93 ± 0.05) compared to White (4.53 ± 0.05; *p* < 0.0001). However, there were practical differences in FCR between stocking densities within a strain that may equate to cost differences in production; FCR was improved for both strains at the LSD. Brown FCR was 3.97 and 3.90 kg of feed to kg of body weight for HSD and LSD, respectively. For White FCR, HSD and LSD were 4.56 and 4.49 kg of feed to kg of body weight, respectively. 

Mortality remained low throughout this study, with an average of 1.02 ± 0.24% mortality from 2 to 16 WOA. Brown had higher mortality compared to White, 1.6% versus 0.4%, respectively (*p* = 0.01).

### 3.3. Blood Parameters

#### 3.3.1. Differential WBC Count

Strain affected percent eosinophils (*p* = 0.003), heterophils (*p* < 0.001), lymphocytes (*p* < 0.001), and monocytes (*p* = 0.01; Table 5). The percentage of heterophils for White was lower than for Brown. Average eosinophils, monocytes, and lymphocytes percentages were higher in White compared to Brown. Age affected the WBC differential count (Table 5) and packed cell volume (PCV; Table 5). WBC components decreased and then increased from month to month, creating a wave-like pattern in cell populations and PCV. Additionally, basophils (*p* = 0.009), lymphocytes (*p* = 0.008), and monocytes (*p* = 0.02) had a stocking density × age interaction effect (Table 5). However, the percentages of lymphocytes, monocytes, and basophils were not different when comparing the HSD and LSD within a given age (*p* > 0.05). 

#### 3.3.2. H:L Ratio, Corticosterone, and Cortisol

The heterophil to lymphocyte (H:L) ratio, a marker of stress, was affected by strain (*p* < 0.0001). H:L ratios were lower in White (Brown: 0.533 ± 0.023; White: 0.225 ± 0.025; *p* < 0.0001). There was also a stocking density × age interaction (*p* = 0.04) for H:L ratios. However, the H:L ratio was not different when comparing the HSD and LSD within a given age (*p* > 0.05).

Plasma cortisol and corticosterone are measures of HPA-axis activation. Cortisol had a significant strain × age interaction effect (*p* < 0.003). Brown had almost double the level of cortisol (Brown: 0.071 ± 0.089 ng/mL, White: 0.038 ± 0.089 ng/mL). However, when comparing cortisol levels at the same ages, strain differences were only significant at 16 WOA for cortisol (Brown: 0.115 ± 0.013, White: 0.047 ± 0.013; *p* = 0.0003). Corticosterone (*p* < 0.04) and cortisol (*p* < 0.0001) levels were affected by age. Cortisol levels increased over time (6 WOA: 0.029 ± 0.089 ng/mL; 16 WOA: 0.081 ± 0.089 ng/mL; *p* = 0.0001), while corticosterone levels decreased over time (6 WOA: 1.618 ± 0.25 ng/mL; 16 WOA: 0.985 ± 0.25 ng/mL; *p* < 0.04). 

### 3.4. Metabolic Changes

Cholesterol, a proxy of increased gluconeogenesis and metabolic activity, had a significant strain × age interaction effect (*p* = 0.02). White had higher cholesterol levels than Brown at all time points, but this was different at 6 WOA only (Table 6; *p* = 0.0002). Average cholesterol levels were 1081 ± 28.7 ng/mL for Brown and 1242 ± 29.4 ng/mL for White. Relative liver weight had a significant strain × age interaction (*p* = 0.01) effect. White had heavier relative liver weights at 1 WOA (*p* = 0.0003) and 16 WOA (*p* = 0.002) (Table 6). Stocking density did not have a significant effect on cholesterol levels or relative liver weight.

### 3.5. Immune Organ Size and Immune Function

There was a stocking density effect on this study’s average of bursal relative weight (*p* < 0.05); LSD had a heavier bursa compared to HSD (3.32% versus 3.08%). Stocking density × age (*p* = 0.04) was significant for bursal dimensions. The HSD had larger bursal dimensions at 1, 3, and 12 WOA (*p* > 0.05). At the last sampling time point, LSD had larger bursal dimensions (*p* = 0.02). There were effects of stocking density × age (*p* = 0.02) on the Newcastle Vaccine Disease (NDV) IgG/IgY antibody titer (Figure 4). HSD generated greater titers at 9, 12, and 16 WOA, but these differences were not statistically significant.

There were effects of strain × age (*p* = 0.0002) on the Newcastle Vaccine Disease (NDV) IgG/IgY antibody titer (Figure 4). White had a higher titer compared to Brown at 6, 9, and 12 WOA (*p* < 0.03). Brown had a higher titer at 16 WOA (*p* > 0.05). 

Bursal dimensions and relative bursal weight had a strain × age interaction effect (*p* < 0.0002; Figure A5). As age increased, so did bursal dimensions. The bursa was 72.3 ± 18.5 mm^2^ at 1 WOA and grew to 426.5 ± 13.2 mm^2^ at 16 WOA. When broken down into weeks, at 1 WOA, all bursometer sizes were 2 or 3; by 16 WOA, approximately 55% of the measured bursas were size 6–8, and approximately 44% of measured bursas were size 4–5. Brown had a smaller bursal size and weight at all time points, except at 1 WOA when Brown had a slightly heavier bursa compared to White (*p* > 0.05). Strain differences were only significant at 16 WOA; bursal dimensions were larger in White (White: 503.1 ± 15.8 mm^2^; Brown: 349.8 ± 15.6 mm^2^; *p* < 0.0001). When evaluating this study’s average bursal size by bursometer, White had larger bursas compared to Brown (Figure A5). Bursal relative weight differences between strains at 6 (White: 0.62 ± 0.02%, Brown: 0.51 ± 0.02%; *p* = 0.01) and 16 WOA (White: 0.27 ± 0.01%, Brown: 0.14 ± 0.01%; *p* < 0.0001) were statistically different. Based on bursometer size, White had higher percentages of larger bursas compared to Brown (Figure 4). There was a strong correlation between bursal dimensions (as measured by ImageJ) and bursometer (r^2^ = 0.8378; *p* < 0.0001). 

As age increased, so did the second thymus lobe dimensions (mm^3^; *p* < 0.0001). At each time point, the second thymus lobe grew larger (*p* < 0.02) in volume, though growth slowed after the first 8 weeks. Similarly, the thymus grew in relative weight until 8 WOA, and then the thymus comprised a smaller percentage of the birds’ body weights at 12 and 16 WOA. On average, the second left thymus lobe was larger than the second right thymus lobe at every time point. White had a larger relative weight of the thymus compared to Brown (White: Right: 0.045 ± 0.001%, Left: 0.060 ± 0.002%, Brown: Right: 0.036 ± 0.001%; Left: 0.051 ± 0.002; *p* < 0.0003). 

Spleen volume had a strain × age interaction effect (Figure A6; *p* < 0.0001). The spleen’s relative weight (%) had age (*p* < 0.0001) and strain × age interaction (*p* < 0.0001) effects. In general, white had a larger volume and relative weight at the first two time points, 1 and 3 WOA. After that point, the trend switched, and Brown had the larger spleen volume and relative weight for the remainder of this study. Strain differences were present at 12 and 16 WOA (*p* < 0.0001) for spleen volume and relative weight (Figure A6). Spleen volume (mm^3^) was higher in Brown at both ages (12 WOA Brown: 5074 ± 169, White: 3981 ± 167; 16 WOA Brown: 5587 ± 147, White: 4501 ± 148). The relative weight of the spleen (%) was higher in Brown than in White at both 12 and 16 WOA (12 WOA Brown: 0.23 ± 0.007, White: 0.21 ± 0.007; At 16 WOA Brown: 0.20 ± 0.006, White: 0.19 ± 0.005).

## 4. Discussion

Management choices during the first 16 weeks of life can have lasting effects on a pullet. In particular, stocking density can affect the birds’ well-being and behaviors [8,10], and both consumers and farmers cite stocking density as one of the most important factors in well-being [12,13]. However, few studies have evaluated stocking density in cage-free pullets. As consumers and businesses desire more cage-free housing, there is a need to understand how pullets adapt to and are affected by different stocking densities in cage-free systems. Our experiment used two strains of birds, Brown and White, housed at either a high-stocking density (HSD) or low-stocking density (LSD), with the HSD representing something commercial farms might use and the LSD representing densities that allowed for the maximization of natural bird behaviors based on computer models [2,17]. Our hypothesis was that birds housed at HSD would have worse bird-based welfare parameters—the measured parameters would have greater alterations from baseline levels. 

In our study, only a few parameters were unaffected by strain, age, or stocking density. The birds had perfect scores, indicating no damage, for footpad scores, keel tip fractures, keel scores, and neck and feather coverage for the study duration. Therefore, these parameters may be less useful to measure in future pullet stress and welfare studies. However, the remainder of the parameters tested in our study were affected by either strain, age, stocking density, or an interaction effect between these factors.

When evaluating biomarkers in blood, there were primarily age and strain effects. Components of the white blood cell differential count increased and decreased over time. The same tagged birds were followed for the duration of this study; thus, our observed changes are less likely the result of inter-bird variation alone. Instead, these cell-population changes may be a normal pattern during development. Heterophilia and lymphopenia are common during the first few days of life in chicks [47], so perhaps the ebb and flow of the other cell populations are representative of an additional age-related pattern. 

Another biomarker affected by age was the packed cell volume (PCV). PCV is a marker of dehydration where elevated levels indicate increased dehydration [48], and some studies have found correlations between increased PCV and elevated levels of stress [29,49]. In our study, PCV was elevated at 16 WOA compared to 9 and 12 WOA, and Brown was higher than White. This could be due to stress, elevated environmental temperatures (94 °F at 16 WOA), or normal age changes. It is unclear if the differences in PCV between strains represent normal variation or that one strain could not cope with the stocking density or heat stressor. Because there were no significant differences between strains at 4, 8, or 12 WOA, “normal” PCV was likely relatively similar between strains, but that might not hold true as the birds grew older. 

Similar to the strain effects in PCV, the WBC differential count and resulting H:L ratio showed distinct differences between strains. H:L ratios are made up of two populations—heterophils and lymphocytes—and are often used as an indirect measure of stress in birds in the absence of immune system activation. In Brown, the proportions of heterophils were increased, and the proportions of lymphocytes were decreased, while in White, the opposite effects were recorded. This explains why Brown had higher H:L ratios (0.5) compared to White (0.2). These differences may indicate a higher baseline level of stress in Brown or purely natural variation between different genetic strains or within chicken species. Some studies suggest that H:L ratios above 1.2 may indicate a stressful situation [50,51] and that values below 0.6 indicate low environmental stress [50]. This would indicate that in our study, the birds fell below a threshold for stress and that neither stocking density used triggered a stress response as measured by H:L ratio. Despite these guidelines, the utility of these thresholds is questionable at best because H:L ratio interpretations are limited by individual versus population differences [52], and these measurements are only a snapshot in time. Arguably, the reliability of H:L ratios should be questioned because while the H:L ratio has been used extensively as a measure of stress [35,46,53,54], one study described an 89.7% correlation between plasma corticosterone concentrations and H:L ratios [55], some stressors do not solicit a change in H:L ratio [28,34,56], and the specific mechanisms that lead to elevated H:L ratios during stress are not fully elucidated [46,57]. Thus, further investigation into the H:L ratio and its utility as a measure of stress should be evaluated.

In addition to H:L ratios, glucocorticoid levels are frequently used to measure stress in avian species. In our study, while unaffected by stocking density, baseline corticosterone, and cortisol plasma levels were affected by age. In our study, as the birds grew, cortisol levels increased, and corticosterone levels decreased. This is opposite to the trends seen in wild birds. Corticosterone increased with age in wild songbirds [58,59]. However, there are no specific reference ranges of baseline corticosterone and cortisol levels in chickens. Corticosterone levels in our study ranged from 0.13 to 5.6 ng/mL, representing both stressed and unstressed birds based on previous research. Past reports demonstrated corticosterone concentrations from <0.3 to 20+ ng/mL in unstressed chickens and 0.25–150 ng/mL in stressed chickens [34]. With overlapping reference ranges for stressed and unstressed chickens, interpretation of corticosterone levels becomes limited. The wide reference ranges may be partially due to the limitations of antibody-based assays. Additionally, the interpretation of corticosterone and cortisol is further complicated by the phases of HPA-axis activation. With stocking density, as the birds grow, the expectation is that stress is increased and becomes chronic, which can elevate corticosterone and cortisol levels [60,61]. However, if a stressor is experienced for too long, like a stocking density stressor for 16 weeks, the HPA axis can become desensitized and can no longer respond [62]. The desensitization of the HPA could result in near baseline levels of glucocorticoids, which do not represent the true stress levels of the birds. The sampling milieu of our study only provides a snapshot of what glucocorticoid levels are at any given moment. Thus, it is unclear whether the patterns observed in our study are normal, expected differences, or represent differences in bird welfare with chronic stress. The results further illustrate a need to investigate the role of both corticosterone and cortisol during periods of stress.

Though glucocorticoids are often the measurement of choice for detecting stress in animals, immunosuppression is also a commonly cited measurement of stress [57]. Immunosuppression or reduced functionality can be measured through decreased immune organ size or decreased vaccine titers [35,57,61,63,64]. In our study, only relative bursal weight (%) was affected by stocking density alone. Relative bursal weights were heavier in the birds housed at LSD compared to HSD. Additionally, bursal dimensions were increased in LSD at 16 WOA compared to HSD. However, most organ measurements were unaffected by stocking density. Thus, further research is needed to identify if altered bursal weight affected functionality because there were no stocking density effects at a given age on the Newcastle vaccine titers. 

While most biomarkers of immunity were unaffected by stocking density, they were affected by age and strain. Both thymus and bursal size increased until the termination of this study. Past studies have debated the exact timing of bursal regression, suggesting everything from 24 days of age [65] to 4–6 months of age [66,67,68]. Because the mean bursal size was still increasing at the termination of our study, the regression of the bursa may occur at an even later age than previously described. The bursal sizes reported in this study generate a guideline for normal sizes as laying hen pullets grow. Because bursal size is less frequently evaluated in pullets compared to broilers and because general size guidelines exist for broilers but not laying hen pullets, the results of this study may help differentiate normal bursal growth from disease or stressful conditions. This may be made easier by using a bursometer. There was a strong correlation between bursal dimensions (as measured by ImageJ software, Version 1.52t 30) and bursometer measurements, indicating that the bursometer may be an accurate tool in the field to approximate bursal size changes. 

Similar to bursal age changes, the thymus is thought to regress between 12 and 16 WOA [64,69]. In this study, both the second lobes of the left and right thymus increased in size for the duration of this study, indicating that, like the bursa, the thymus may not regress until after 16 WOA. However, caution in the interpretation of thymus size changes should be taken as only the second lobe on each side was measured, and the entire thymus consists of 7–8 lobes on each side. Additionally, like in previous reports [64], the left side was larger than the right, which might affect interpretation. 

In addition to age effects, immune organ size was also affected by strain. The spleen size was greater in Brown at 12 and 16 WOA, but the bursal size at 16 WOA and relative weight of the bursa at both 6 and 16 WOA was larger in White compared to Brown. The relative weights of the thymus were also greater in White compared to Brown. Differences between strains are not unexpected, as previous research has reported that relative bursal size varies widely between types of birds [64,70,71], and past research has demonstrated that differences in bursal size are partially attributed to strain [67]. The thymus and spleen are likely no exception. The differences in immune organ size between strains may have also equated to differences in immune functions. The Newcastle IgG/IgY titer had a strain × age effect, with White having higher titers at 6, 9, and 12 WOA compared to Brown. Future studies should evaluate correlations between primary lymphoid organ measurements and adaptive immune function.

Stress can have widespread effects, not only altering immunity but metabolism and growth as well. Cholesterol and liver size may serve as proxies for increased metabolic processes. During stressful conditions, the body ramps up gluconeogenesis and lipogenesis, which increases fat deposition in the liver and adipose tissues of chickens [31,34,35,72]. Thus, during periods of stress, liver size, and circulating cholesterol levels may increase [55,61,73]. In our study, however, strain and age had a greater influence on liver size and cholesterol levels than the stocking density stressor. White had larger livers at both 1 and 16 WOA (*p* < 0.0003) and had higher cholesterol levels at 6 WOA (*p* = 0.0002) compared to Brown. Do these levels represent differences in baseline stress and metabolism or normal, natural variations between strains? If larger livers and higher cholesterol indicate higher baseline stress levels, the reason for the differences in timelines between liver size and cholesterol levels (1 and 16 versus 6 WOA) is unclear. Further research into normal values among strains needs to be completed to truly understand the cause of these differences.

While liver size and cholesterol were unaffected by stocking density, other metabolic biomarkers like body weight, body weight uniformity, and FCR were affected by stocking density. Body weight was the only parameter with a significant three-way interaction effect of strain, stocking density, and age. However, within a strain and age (e.g., Brown at 6 WOA, White at 8 WOA, etc.), there were no differences between the high and low-stocking densities. There was an effect of strain and age in that Brown was heavier than White from 7 to 16 WOA (*p* < 0.003). This is to be expected as the management guidelines for each strain estimate heavier birds for Brown compared to White [36]. Past research has shown differences in growth between strains housed at the same stocking densities and the differential effects of stocking densities on strains. In caged studies of pullets provided 247, 270, 299, or 335 cm^2^/bird, shaver pullets had higher body weights at the low-stocking densities, while LSL-Lite, Lohmann Brown, and Dekalb pullets weighed less when provided 299 cm^2^/bird compared to the other densities, and Lohmann Brown pullets had the highest body weights at the highest and lowest stocking densities [5]. The reason for these strain differences is unclear, but the difference is readily apparent. 

Both body weight uniformity and FCR were affected by stocking density. FCR differences between stocking densities within a strain were not significant, but LSD generated numerically and economically improved FCR for both White and Brown, equating to approximately 0.07 kg less feed per kg of body weight for Brown and White. But while an important component of FCR, feed consumption in our study was unaffected by stocking density. This was an unexpected result as several past studies of caged layers observed decreased feed consumption at high-stocking densities [74,75,76], and a few studies of pullets detected decreased, better FCR at high-stocking densities [5,77] and decreased feed consumption at higher stocking densities [78]. Theoretically, if a bird felt more stress at a certain stocking density, more energy would be put toward maintenance and coping with a stressor instead of growth, as suggested by Johns et al. [79]. Thus, it would be expected that in our study, birds were less stressed at LSD. The reason for the differences between past research and our study is unclear but may be related to the birds’ genetics and behaviors or the unrestricted access to feeders in our study. 

In addition to stocking density effects, FCR was affected by strain, with Brown having an improved conversion rate. This disparity was likely due to differences in genetic efficiencies between Brown and White because total feed consumption was unaffected by strain, and Brown was heavier than White. White may have been more active, which would have caused a poorer FCR compared to Brown. Increased activity means decreased energy put toward weight gain, so behavioral observations would be required to investigate this possibility. 

Unlike FCR, stocking density affected uniformity in different ways for Brown versus White. Brown had better uniformity in LSD, while White had better uniformity in HSD (*p* < 0.02). Though not statistically significant, body weight uniformity was also affected by strain. Uniformity was better in Brown for the first half of this study and better in White for much of the second half of this study. This could equate to practical differences on farms. Better body weight uniformity can create improved FCR, more even egg production, and more even egg size when birds enter production. After the first 7 weeks of this study, Brown weights increased more quickly than White weights. This generated a larger but less uniform bird. 

Bird and flock uniformity is important for uniform egg size and feed efficiency. In addition to body weight, bird frame size, and overall uniformity can be estimated using shank length. Shank length is reported to be a measure of skeletal size and development [32,44]. This study showed a strong correlation between shank length and body weight, indicating that shank length was a good proxy for growth. However, shank length plateaued after 12 WOA, so shank length may not be a good estimate of growth after this point. This plateau also illustrates that frame size and skeletal growth may not occur after 12 WOA, so emphasis on skeletal integrity may be more important during the first 3 months of life. In our study, while Brown was heavier, White appeared to have a larger skeleton as measured by shank length. White had longer shank length at 8 and 12 WOA than Brown (*p* < 0.0001). Thus, White should theoretically have a larger frame with reduced body weight.

In addition to bird size, the overall appearance of the birds in our study was affected by age, strain, and stocking density. Total feather coverage was improved in LSD for Brown and White, which is consistent with the past literature. One theory posits that birds receive more stimuli from other birds and their feathers relative to the litter area when housed at higher stocking densities [22]. These greater stimuli can equate to greater feather pecking and damage. The majority of the past literature described an increase in feather pecking at higher stocking densities [14,26,74,80,81,82,83,84]. However, in our study, White LSD birds had worse tail scores. While past research has found that pecking at tail feathers occurred for longer bouts than any other body region [85], the reason why this was exacerbated in LSD for White only is unclear.

In addition to stocking density effects, feather coverage was affected by age, strain, and a strain × age interaction effect. Feather coverage varied from month to month and was likely the result of frequent molting during the pullet phase. Another consideration is that pecking behaviors change as the birds grow [86]. A study of Thornber and Shaver strains found no differences between strains during the first 8 weeks of life; then, the Shaver strains pecked more until 12 WOA, and from 13–18 WOA, Thornbers pecked more [19]. In our study, similar age effects of pecking were seen. White had a better tail and total feather coverage compared to Brown at 8 and 12 WOA, but at 16 WOA, Brown had better coverage, indicating changes in feather pecking behaviors. As a whole, White had less severe scores or less damage to the back and wings. This is in opposition to previous studies, which found that white strains tended to peck at the back more than brown strains [85] and that white strains had a higher incidence of severe feather pecking [87]. Some of the discrepancies may be due to strain variances or how the feathers are visualized. In our study, starting around 13 WOA, severe feather pecking was noted, and light levels were reduced. Several past studies have found strain differences in feather pecking behavior and feather condition. In general, adult brown strains tended to peck at feathers more frequently than white strains [85,87]. But an opposite effect was seen in pullets, where LSL-lite and Dekalb pullets (white strains) feather pecked more than Lohmann brown pullets [5]. Feather pecking may have as high of heritability as 0.09–1.04 [88], so strain and the pecking genes associated with each strain could affect plumage scores. This may partially explain the differences between our study and past research with other strains. The marking system used may also have contributed to some of the feather damage seen. Color was placed on the shoulder and back regions, and wing and back feather coverages were the poorest at 8 and 12 WOA, respectively. 

While severe feather pecking could increase mortality levels, mortality in our study was low and was not affected by feather-pecking behaviors. There were strain effects with mean mortality in Brown 1.2% higher than in White. In total, 29 birds died during this study due to unrelated causes. This relatively small amount of mortality might be purely coincidental or could relate back to overall strain differences.

In summary, stocking density alone only affected the relative bursa of Fabricius weight. Birds housed at LSD had larger relative bursal weights (%) compared to HSD. The implications of this difference are unclear because the other immune organs and vaccine titers were not affected by stocking density alone. Of the stocking density × strain interaction effects, Brown birds’ body weight uniformity and feather coverage were worse at the higher stocking density, while the opposite was true for White. The feed conversion ratio (FCR) was improved at LSD for both strains. The majority of parameters assessed in our study were affected by strain and age only. These observations suggest that the parameters were sensitive enough to detect change, but that relatively few changes were seen because of stocking density effects.

The results of this study provide insight into differences in modern genetic strains and generate a rough reference range for future studies and for evaluation in the field. However, these baseline values may only be valid for Lohmann LB-Lite and Lohmann LSL-Lite, and extrapolation to other strains would be limited, as evidenced by the strain differences observed in our study. Likely, data from the Lohmann LB-Lite would be more useful for other brown strains, and data from the Lohmann LSL-Lite for other white strains. However, without further knowledge, this should not be assumed. The utility of the bird-based welfare parameters tested in this study is uncertain because there were no clear-cut differences between the two stocking densities tested. Most parameters in this study were not affected by stocking density but revealed strain and age effects. This suggests that the parameters tested are sensitive enough to detect changes but that few of the changes seen were due to a stocking density stressor. The stocking density stressor may not have been extreme enough compared to adult hens; pullets may be more resistant to stress, or the measurements used may not be effective measures of stress and welfare. There is a limited understanding of the HPA axis in avian species, and this study illustrates a need to further understand downstream effects and triggers for HPA axis activation. 

## 5. Conclusions

This study provides insight into how two of the modern genetic strains of laying hen pullets differ, generating reference ranges for physiological, immunological, and physical characteristics. The significant strain differences within this study outline the need for future studies specific to individual strains of pullets. Of the multitude of bird-based welfare parameters evaluated in this study, there were no consistent, definitive effects of stocking density. Most parameters evaluated were not affected by stocking density but were affected by both the strain and age of the bird. The stocking densities used may not have been extreme enough; the strains of pullets used may be resistant to stress, or the parameters evaluated may not be accurate measurements of stress and welfare. This study demonstrates a clear need for further research into the HPA axis, stress, and welfare in avian species, particularly in laying hens.

## Figures and Tables

**Figure 1 animals-14-01513-f001:**
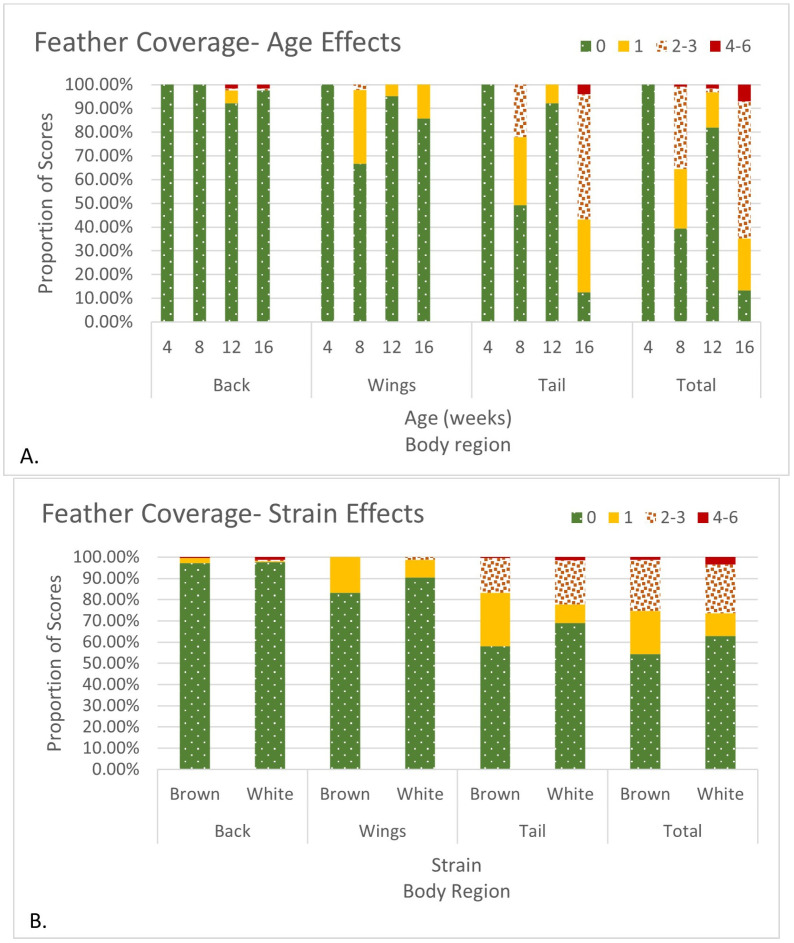
Age (**A**) and Strain (**B**) effects on feather coverage. Feather coverage is based on past studies [39,40] and is a scale of increasing severity where for individual body regions, scores of 0 are perfect, and scores above 3 indicate tissue damage. The maximum score achieved by body region was a 4 for back and wings and 5 for tail. Total feather coverage scores are a sum of all body region feather coverage scores (0 = no damage; 30 = worst possible damage). The maximum score achieved for total feather coverage was 6 for Lohmann LSL-Lite (White) or Lohmann LB-Lite (Brown) pullets housed on the floor from 0 to 16 weeks of age.

**Figure 2 animals-14-01513-f002:**
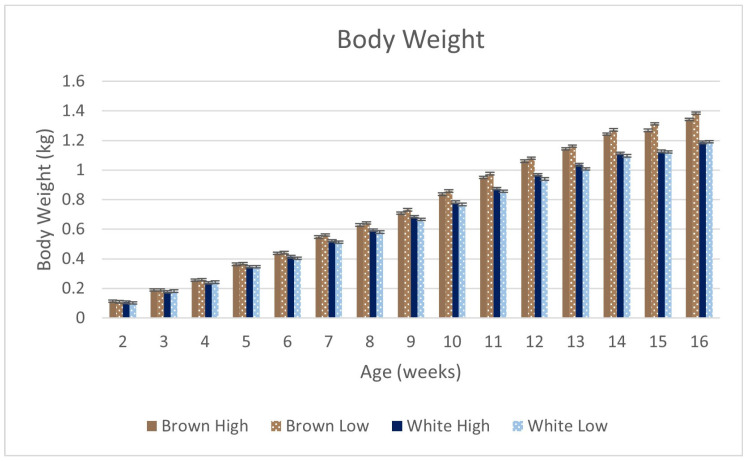
Strain × density × age effects for body weights from 2 to 16 WOA of Lohmann LB-Lite (Brown) Lohmann LSL-Lite (White) pullets housed on the floor. There were strain (*p* < 0.0001), age (*p* < 0.0001), strain × density (*p* < 0.04), strain × age (*p* < 0.0001), density × age (*p* < 0.001), and strain × density × age (*p* < 0.0001) effects on body weight. Density had no effect when evaluating the same strain at the same age.

**Figure 3 animals-14-01513-f003:**
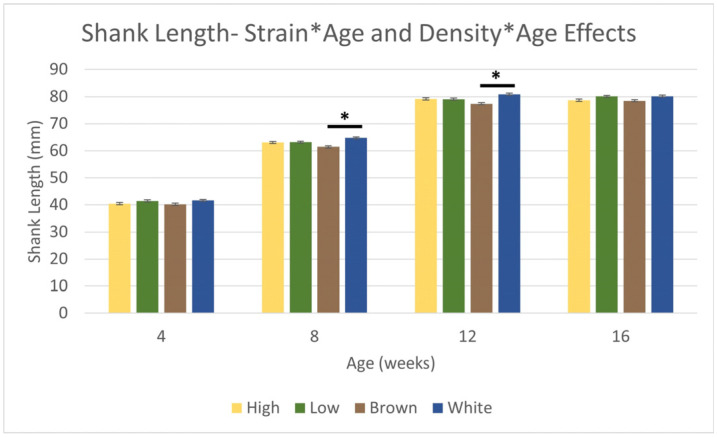
There were strain (*p* < 0.0001), age (*p* < 0.0001), stocking density × age (*p* < 0.05), and strain × age (*p* < 0.0002) effects on shank length of Lohmann LB-Lite (Brown) and Lohmann LSL-Lite (White) pullets. Asterisks denote differences within an age group (*p* < 0.05); within an age group, there were no differences between stocking densities (*p* > 0.05).

**Figure 4 animals-14-01513-f004:**
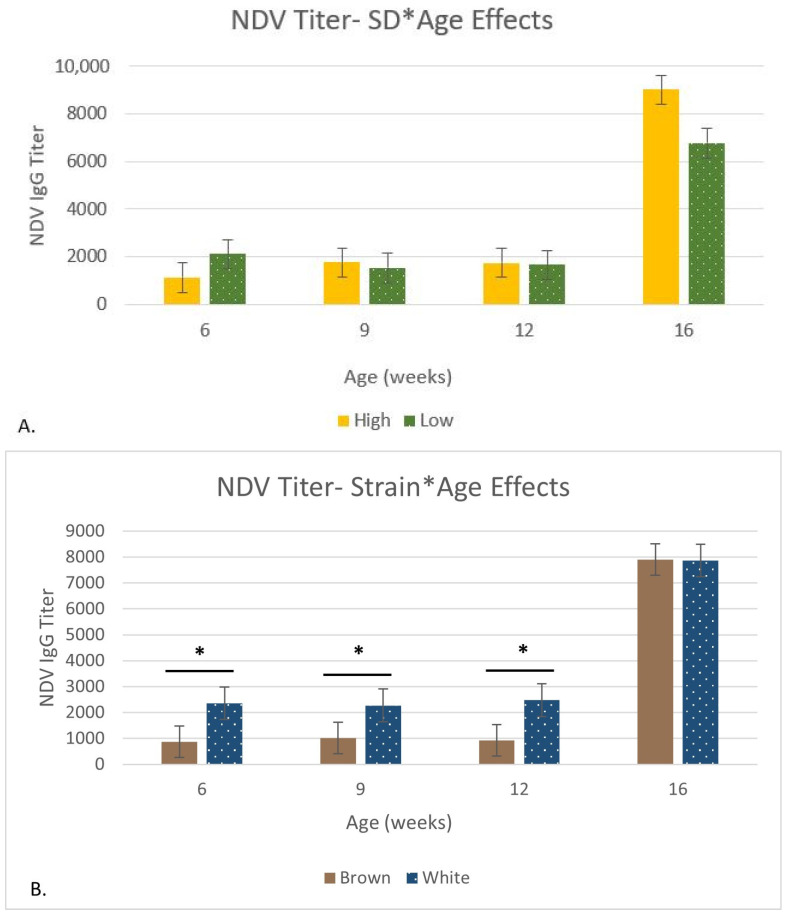
There were strain (*p* < 0.0001), age (*p* < 0.0001), density × age (*p* < 0.02; (**A**)), and strain × age (*p* = 0.0002; (**B**)) effects on Newcastle Disease Virus IgG antibody levels. Within an age group, no differences between high and low-stocking densities (*p* > 0.05, (**A**)) were present. For the strain × age interaction, asterisks denote differences within an age group (*p* < 0.05, (**B**)). On day 13, birds received Newcastle as a coarse spray and again on days 33 and 56 as a coarse spray. Newcastle was delivered as a breast injection on day 84. Blood was collected on days 40, 61, 81, and 110.

**Table 1 animals-14-01513-t001:** Approximate resource space per chick due to natural and sampling mortality, resulting in slight variation from pen to pen ^1^.

	2–3 WOA	3–6 WOA	6–12 WOA	12–16 WOA	Target
Area (cm^2^)					
-HSD	619.1	635.2	652.2	670.2	619.1
-LSD	1248.9	1281.5	1315.8	1352.0	1248.9
Feeder space (cm)	7.1	7.4	7.6	7.8	7.1
Drinker space (cm)	3.5	3.7	3.8	3.9	3.5

^1^ HSD = high-stocking density; LSD = low-stocking density.

**Table 2 animals-14-01513-t002:** Welfare Assessment of Pullets.

Feather Coverage Scoring System ^1^	
Score	Description
0	Completely covered
1	<10% of feathers missing, damaged, or broken
2	11–50% of feathers missing, damaged, or broken (no tissue damage)
3	>50% of feathers missing, damaged, or broken (no tissue damage)
4	10–50% of feathers missing, damaged, or broken (tissue damage present)
5	>51% of feathers missing, damaged, or broken (tissue damage present)
Keel Scoring System ^2^	
Score	Description
0	No deviations, deformations or thickened sections; keel bone completely straight
1	Tip fracture and/or deviation (≤2 cm)
2	Multiple or more severe deviation or deformation of keel bone (including thickened sections; >2 cm)
Keel Tip Fracture ^2^	
Yes	Keel tip fracture present
No	Keel tip fracture absent
Footpad Scoring System ^2^	
0	Feet intact; no or minimal proliferation of epithelium
1	Necrosis or proliferation of epithelium or chronic bumble foot with no or moderate swelling
2	Swollen (dorsally visible)
Shank length	
Measured from the hock to the footpad as a proxy for skeletal size and growth ^3^	

^1^ Adapted from Morrisey et al. [39] and Arrazola et al. [40]. The neck, back, tail, belly/vent region, wings, and legs are evaluated. A total score is generated from the sum of these scores, where a score of 0 is perfect, and a score of 30 is the worst possible score. ^2^ Based on the Welfare Quality^®^ Assessment (2009). ^3^ [42,43,44].

**Table 3 animals-14-01513-t003:** Statistical Models Used.

	Parameter	Model
Fixed Linear	% Eosinophils and Heterophils, Thymus Volume and Relative Weight, Corticosterone, Cortisol	Y_ijk_ = μ + ST_i_ + SD_j_ + A_k_ + ε_ijk_
	FCR, Mortality	Y_ijk_ = μ + ST_i_ + SD_j_ + ε_ijk_
	H:L ratio, % Lymphocytes and Monocytes	Y_ijk_ = μ + ST_i_ + SD_j_ + A_k_ + SD_j_ × A_k_ +ε_ijk_
	Relative Bursal and Liver Weight	Y_ijk_ = μ + ST_i_ + SD_j_ + A_k_ + ST_i_ × A_k_ + ε_ijk_
	Uniformity	Y_ijk_ = μ + ST_i_ + SD_j_ + A_k_ + ST_i_ × SD_j_ + ST_i_ × A_k_ + ε_ijk_
Linear Mixed	PCV	Y_ijk_ = μ + ST_i_ + SD_j_ + A_k_ + ST_i_ × A_k_ + B + ε_ijk_
	Body Weight	Y_ijk_ = μ + ST_i_ + SD_j_ + A_k_ + ST_i_ × SD_j_ + ST_i_ × A_k_ + SD_j_ × A_k_ + B + ε_ijk_
	Shank Length	Y_ijk_ = μ + ST_i_ + SD_j_ + A_k_ + SD_j_ × A_k_ + P(B) + ε_ijk_
	% Basophils	Y_ijk_ = μ + ST_i_ + SD_j_ + A_k_ + SD_j_ × A_k_ + P + ε_ijk_
	Cholesterol	Y_ijk_ = μ + ST_i_ + SD_j_ + A_k_ + ST_i_ × A_k_ + P + ε_ijk_
	NDV Titer	Y_ijk_ = μ + ST_i_ + SD_j_ + A_k_ + ST_i_ × A_k_ + SD_j_ × A_k_ + P + ε_ijk_
	Relative splenic weight	Y_ijk_ = μ + ST_i_ + SD_j_ + A_k_ + ST_i_ × A_k_ + P + ε_ijk_
	Spleen volume	Y_ijk_ = μ + ST_i_ + SD_j_ + A_k_ + ST_i_ × A_k_ + R(P) + ε_ijk_
	Bursal Dimensions	Y_ijk_ = μ + ST_i_ + SD_j_ + A_k_ + ST_i_ × A_k_ + SD_j_ × A_k_ + R + ε_ijk_
Multinomial Logistic Regression	Bursometer size, Back feather coverage, Wing feather coverage	Y_ijk_ = μ + ST_i_ + A_k_ + ε_ijk_
	Tail feather coverage, Total feather coverage	Y_ijk_ = μ + ST_i_ × SD_j_ + ST_i_ × A_k_ + ε_ijk_

Parameter Definitions: μ = model constant ST_i_ = Fixed effect of strain (i = 1 to 2) SD_j_ = Fixed effect of stocking density (j = 1 to 2) A_k_ = Fixed effect of age (k = 1 to 4) SD_j_ × A_k_ = Interaction effect of stocking density and age (jk = 1–8) ST_i_ × A_k_ = Interaction effect of strain and age (ik = 1–8) ST_i_ × SD_j_ = Interaction effect of strain and stocking density (ij = 1–4) B = Random effect of bird P = Random effect of pen R = Random effect of room P(B) = Random effect of bird nested within pen R(P) = Random effect of pen nested within room ε_ijk_ = Random error variation.

**Table 4 animals-14-01513-t004:** Tail and Total Feather Coverage Scores. Feather scores of two different strains of laying hen at high and low-stocking densities during the pullet phase (0–16 WOA) ^1^. Numbers presented are percentages of birds exhibiting a given feather score at a given age or as an average of this study. The maximum score per body region is 6; the maximum total body feather coverage score is 30. In this study, tail feather coverage never exceeded score of 5. * The highest total feather coverage score achieved in this study, as a sum of all body regions, was 6.

(A) Tail Feather Coverage Score							
Brown Strain							
Density	0	1	2		3	4	5
High	56.25%	25.78%	17.19%		0.78%	0.00%	0.00%
Low	59.84%	24.41%	14.96%		0.00%	0.79%	0.00%
Age (weeks)	0	1	2		3	4	5
4	100.00%	0.00%	0.00%		0.00%	0.00%	0.00%
8	18.75%	39.06%	42.19%		0.00%	0.00%	0.00%
12	87.50%	12.50%	0.00%		0.00%	0.00%	0.00%
16	25.40%	49.21%	22.22%		1.59%	1.59%	0.00%
White Strain							
Density	0	1	2		3	4	5
High	69.63%	11.85%	16.30%		0.74%	0.74%	0.74%
Low	68.25%	5.56%	19.84%		4.76%	0.79%	0.79%
Age (weeks)	0	1	2		3	4	5
4	100.00%	0.00%	0.00%		0.00%	0.00%	0.00%
8	77.94%	19.12%	2.94%		0.00%	0.00%	0.00%
12	96.88%	3.13%	0.00%		0.00%	0.00%	0.00%
16	0.00%	12.50%	70.31%		10.94%	3.13%	3.13%
(B) Total Feather Coverage Score							
Brown Strain							
Density	0	1	2	3	4	5	6 *
High	51.56%	21.88%	20.31%	5.47%	0.78%	0.00%	0.00%
Low	57.03%	18.75%	15.63%	7.03%	0.00%	1.56%	0.00%
Age (weeks)	0	1	2	3	4	5	6
4	100.00%	0.00%	0.00%	0.00%	0.00%	0.00%	0.00%
8	18.75%	23.44%	40.63%	17.19%	0.00%	0.00%	0.00%
12	71.88%	26.56%	1.56%	0.00%	0.00%	0.00%	0.00%
16	26.56%	31.25%	29.69%	7.81%	1.56%	3.13%	0.00%
White Strain							
Density	0	1	2	3	4	5	6
High	65.44%	12.50%	15.44%	2.94%	1.47%	1.47%	0.74%
Low	60.32%	8.73%	21.43%	6.35%	2.38%	0.79%	0.00%
Age (weeks)	0	1	2	3	4	5	6
4	100.00%	0.00%	0.00%	0.00%	0.00%	0.00%	0.00%
8	58.82%	26.47%	8.82%	4.41%	1.47%	0.00%	0.00%
12	92.19%	3.13%	0.00%	1.56%	1.56%	1.56%	0.00%
16	0.00%	12.50%	65.63%	12.50%	4.69%	3.13%	1.56%

^1^ Feather coverage is based on past studies [39,40], and is a scale of increasing severity, where for individual body regions scores of 0 are perfect, and scores above 3 indicate tissue damage. Total feather coverage scores are a sum of all body region feather coverage scores (0 = no damage, 30 = worst possible damage). The maximum total feather coverage score achieved in this study was 6.

**Table 5 animals-14-01513-t005:** Differential WBC Count ^1^ and PCV of Lohmann LB-Lite (Brown) Lohmann LSL-Lite (White) pullets housed on the floor at a high or low-stocking density from 2 to 16 WOA ^2^.

Age Effects								
Age (Weeks)	PCV	Heterophils	Lymphocytes	Basophils		Eosinophils		Monocytes
6	--	22.3 ± 1.11 ^A^	68.3 ± 1.34 ^A^	2.40 ± 0.20 ^A,B^		1.81 ± 0.18 ^A,B^		5.30 ± 0.29 ^A^
9	29.1 ± 0.25 ^B^	21.2 ± 1.08 ^A^	72.7 ± 1.31 ^A^	2.01 ± 0.20 ^B^		1.39 ± 0.18 ^B^		2.94 ± 0.28 ^B^
12	28.5 ± 0.24 ^B^	20.9 ± 1.05 ^A^	69.9 ± 1.27 ^A^	2.85 ± 0.19 ^A^		2.27 ± 0.17 ^A^		4.17 ± 0.28 ^C^
16	30.5 ± 0.23 ^A^	22.3 ± 1.02 ^A^	69.6 ± 1.23 ^A^	2.56 ± 0.19 ^A,B^		1.88 ± 0.17 ^A,B^		3.26 ± 0.27 ^B,C^
*p*-value	<0.001	>0.05	>0.05	0.01		0.006		0.001
Strain Effects								
	PCV	Heterophils	Lymphocytes	Basophils		Eosinophils		Monocytes
Brown	29.7 ± 0.22 ^A^	28.6 ± 0.73 ^A^	63.5 ± 0.88 ^B^	2.56 ± 0.16 ^A^		1.57 ± 0.12 ^B^		3.56 ± 0.19 ^B^
White	29.1 ± 0.24 ^A^	14.8 ± 0.78 ^B^	76.7 ± 0.94 ^A^	2.35 ± 0.16 ^A^		2.11 ± 0.13 ^A^		4.27 ± 0.20 ^A^
*p*-value	>0.05	<0.001	<0.001	>0.05		0.003		0.01
Strain × age Effects								
	PCV							
Age (weeks) /Strain	Brown	White						
6	--	--						
9	29.3 ± 0.34 ^B,a^	29.0 ± 0.37 ^A,a^						
12	28.4 ± 0.32 ^B,a^	28.6 ± 0.34 ^A,a^						
16	31.4 ± 0.32 ^A,b^	29.6 ± 0.34 ^A,a^						
*p*-value	0.004							
Density × age Effects								
Age (weeks) Density	Heterophils	Lymphocytes	Basophils		Eosinophils		Monocytes	
6 High Low	22.0 ± 1.23 ^A^ 22.6 ± 1.23 ^A^	70.5 ± 1.90 ^A,B^ 66.0 ± 1.90 ^B^	2.32 ± 0.29 ^A,B^ 2.49 ± 0.29 ^A,B^		1.80 ± 0.20 ^A,B,C,D^ 1.82 ± 0.20 ^A,B,C,D^		4.89 ± 0.41 ^A,C^ 5.70 ± 0.41 ^A^	
9 *High* *Low*	20.9 ± 1.21 ^A^ 21.6 ± 1.20 ^A^	71.0 ± 1.87 ^A,B^ 74.3 ± 1.84 ^A^	2.57 ± 0.28 ^A,B^ 1.45 ± 0.28 ^B^		1.38 ± 0.20 ^B,D^ 1.40 ± 0.20 ^C,D^		3.54 ± 0.41 ^B,C^ 2.33 ± 0.40 ^B,D^	
12 *High* *Low*	20.6 ± 1.19 ^A^ 21.2 ± 1.16 ^A^	72.0 ± 1.84 ^A,B^ 67.7 ± 1.74 ^A,B^	2.64 ± 0.28 ^A,B^ 3.06 ± 0.26 ^A^		2.26 ± 0.20 ^C,A^ 2.28 ± 0.19 ^A,B^		4.17 ± 0.40 ^A,C,E^ 4.18 ± 0.38 ^A,C,E^	
16 *High* *Low*	22.0 ± 1.15 ^A^ 22.7 ± 1.14 ^A^	66.9 ± 1.77 ^A,B^ 72.3 ± 1.72 ^A,B^	2.89 ± 0.27 ^A^ 2.22 ± 0.26 ^A,B^		1.87 ± 0.19 ^A,B,C,D^ 1.89 ± 0.19 ^A,B,C,D^		3.97 ± 0.38 ^A,C,D,F^ 2.54 ± 0.37 ^B,E,F^	
*p*-value	>0.05	0.008	0.009		>0.05		0.02	

^1^ Cell types are percentages of a 100-cell differential count. ^2^ For main effects, ages or strains that do not share a letter are different from each other (*p* < 0.05). For age × strain effects, within a strain, ages that do not share upper case letters are different from each other (*p* < 0.05), and within an age, strains that do not share a lower case letter are different from each other (*p* < 0.05). For density × age effects, within a column, groups that do not share a letter are different from each other (*p* < 0.05). No components of the differential WBC count were affected by a strain × age effect and are not presented here. PCV was not affected by density × age effect.

**Table 6 animals-14-01513-t006:** Metabolic changes in Lohmann LB-Lite (Brown) Lohmann LSL-Lite (White) pullets housed on the floor at a high or low-stocking density from 2 to 16 WOA ^1^.

*Age Effects*					
Age (weeks)	Cholesterol (ng/mL)		Age (weeks)	Liver Relative Weight (%)	
			1	4.54 ± 0.05 ^A^	
6	1260 ± 32.3 ^A,C^		3	3.25 ± 0.04 ^B^	
9	1140 ± 32.2 ^B,D^		6	2.80 ± 0.05 ^C^	
12	1170 ± 32.2 ^C,D^		12	2.48 ± 0.04 ^D^	
16	1076 ± 32.2 ^B,D^		16	1.96 ± 0.03 ^E^	
*p*-value	0.0001			<0.0001	
*Strain × age Effects*					
Age (weeks)	Cholesterol (ng/mL)		Age (weeks)	Liver Relative Weight (%)	
	Brown	White		Brown	White
			1	4.30 ± 0.07 ^A,b^	4.79 ± 0.08 ^A,a^
6	1111 ± 45.0 ^A,b^	1409 ± 46.4 ^A,a^	3	3.22 ± 0.05 ^B,a^	3.28 ± 0.05 ^B,a^
9	1052 ± 44.6 ^A,a^	1228 ± 46.4 ^A,C,a^	6	2.76 ± 0.08 ^C,a^	2.84 ± 0.08 ^C,a^
12	1138 ± 44.6 ^A,a^	1202 ± 46.4 ^B,C,a^	12	2.36 ± 0.05 ^D,a^	2.59 ± 0.05 ^C,a^
16	1024 ± 44.6 ^A,a^	1128 ± 46.4 ^B,C,a^	16	1.83 ± 0.05 ^E,b^	2.10 ± 0.04 ^D,a^
*p*-value	0.02			<0.02	

^1^ Within a column, groups with different capital letters are different from each other (*p* < 0.05). Within a row, groups with different lower-case letters are different from each other (*p* < 0.05).

## Data Availability

Datasets are not currently published but can be solicited from the corresponding author.

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
