# Peer review of "Cage-Free Pullets Minimally Affected by Stocking Density Stressors"

_animals, 2024, doi:10.3390/ani14101513_

Round 1
Reviewer 1 Report
Comments and Suggestions for Authors
Dear Authors,
A great deal of effort has been put into the study. However, considering many factors and only two stocking density negatively affected the originality of the study. There is no standard in the expression of results in tables. Tables need to be rearranged. You need to make the hypothesis a little more specific.
Abstract
Line 30: “WOA” ?
Introduction
The amount in which country? Please specify
Materials and Methods
Lines 118-120: It would be more appropriate to write the lighting program in detail.
İnes 126-127: It is necessary to provide the protein, energy, calcium, and phosphorus content of the diet.
Line 130: What are room temperature values and how is the room temperature value adjusted?
Line 136: Please specify how you identified the keel bone scoring.
Results
Why aren't the Table 4 statistical analyses provided?
Table 6: Why is the effect of stocking density on the characteristics specified in Table 6 not given?
H:L ratio is called the best stress response. Why was this value not given?
Figure A6 is not available in the results section.
Author Response
A great deal of effort has been put into the study. However, considering many factors and only two stocking density negatively affected the originality of the study.
The authors agree that the stocking densities themselves are not original. However, the large quantities of parameters evaluated concurrently have never been evaluated prior to this study. Previous stocking density studies have primarily focused on resource-based instead of animal-based welfare measurements and this is the first study of its kind to include a wide variety of both types of welfare measurements and the first of its kind to evaluate these measurements in cage-free pullets.
There is no standard in the expression of results in tables. Tables need to be rearranged.
Thank you for this suggestion. The authors believe that the current arrangement and table setup is the best way to illustrate the data described within the manuscript. Because there is a wide variety of parameters and methodologies used within this study, it is near impossible to have a standardized table layout. Instead, tables have been carefully selected by the authors to provide the best, easily understood illustration of the data.
You need to make the hypothesis a little more specific.
Thank you for pointing out the lack in clarity. In an endeavor to make the hypothesis clearer, additional descriptors have now been added to the hypothesis stated in the introduction (102-105) and discussion (line 430-432).
Abstract- Line 30: “WOA” ?
This acronym (WOA) has now been defined within the abstract.
Introduction- The amount in which country? Please specify
Thank you for drawing attention to this oversight. Country has been specified in line 45.
Materials and Methods
Lines 118-120: It would be more appropriate to write the lighting program in detail.
Thank you for the suggestion, the information has been added to the manuscript (line 123-130).
İnes 126-127: It is necessary to provide the protein, energy, calcium, and phosphorus content of the diet.
The authors appreciate the interest in diet content. A table with nutritional content has been added to the Appendix.
Line 130: What are room temperature values and how is the room temperature value adjusted?
Thank you for drawing this to our attention. Unit of measure and method of temperature adjustment added to the manuscript (line 140-141).
Line 136: Please specify how you identified the keel bone scoring.
Thank you for pointing out this ambiguity. A phrase has been added to line 147 to describe the use of physical palpation for keel scoring.
Results
Why aren't the Table 4 statistical analyses provided?
Thank you for noting this oversight. Tail and total feather coverage statistical models have been added to Table 3.
Table 6: Why is the effect of stocking density on the characteristics specified in Table 6 not given?
Thank you for drawing this to our attention. Stocking density did not have a significant effect on relative liver weight or cholesterol level. A statement detailing this fact has now been added to line 309-310 to limit confusion.
H:L ratio is called the best stress response. Why was this value not given?
The authors would argue that H:L ratios are not the best measurement of stress. There is great debate about what baseline levels of H:L ratios are in chickens. Baseline H:L ratios as well as their relationship to the values found in this study and the debate on accuracy of H:L ratios as a whole is described in lines 456-476.
Figure A6 is not available in the results section.
Figure A6 is described in lines 345-354 and the figure itself is located within the Appendix.
Reviewer 2 Report
Comments and Suggestions for Authors
The study examining the early life of laying hens, which is known as the pullet phase, the focus was on how different rearing conditions affect their development and future productivity. Specifically, the research looked at how varying the amount of space hens had and the breed of the hen influenced their health and egg-laying ability in a cage-free environment, which is less studied compared to traditional caged systems.
In line 36, please add full name of FCR.
In line 199-207, please mention cutoff for statistical significance such as p < 0.05 or p < 0.01.
Please add relevant references In line 514-517.
In line 610-611, please clarify what kind of farm practices.
Author Response
In line 36, please add full name of FCR.
Thank you for drawing attention to this- feed conversion ratio has been written out for clarity (line 36).
In line 199-207, please mention cutoff for statistical significance such as p < 0.05 or p < 0.01.
Line added to the end of this section (line 218) to define statistical significance.
Please add relevant references In line 514-517.
Citations added to illustrate the statements made in 500-503.
In line 610-611, please clarify what kind of farm practices.
We apologize for the confusion in this sentence. The sentence does not imply farm practices as the reviewer states, but rather practical outcomes from farms. We have added information to further explain the practical outcomes that may be possible (line 597-598).
Round 2
Reviewer 1 Report
Comments and Suggestions for Authors
Dear Authors,
Line 36: Delete "study"
Best regards,